# Classification of Urban Agricultural Functional Regions and Their Carbon Effects at the County Level in the Pearl River Delta, China

Zuxuan Song [1], Fangmei Liu [2], Wenbo Lv [2] and Jianwu Yan [2,*]

[1] School of Geography and Planning, Sun Yat-sen University, Guangzhou 510006, China; songzx@mail2.sysu.edu.cn
[2] School of Geography and Tourism, Shaanxi Normal University, Xi'an 710119, China; sybill548@snnu.edu.cn (F.L.); wenbo.lv@snnu.edu.cn (W.L.)
* Correspondence: yanjw@snnu.edu.cn

**Abstract:** Exploring the transformation process of urban agricultural functions and its interaction with carbon effects based on regional differences is of great positive significance for achieving a low-carbon sustainable development of agriculture in metropolitan areas. By using the index system method, self-organizing feature maps (SOFM) network modeling, and Granger causality analysis, we divided the agricultural regional types of the Pearl River Delta (PRD) based on the spatio-temporal changes in urban agricultural functions and carbon effects at the county level in the PRD from 2002 to 2020, and analyzed the carbon effects generated by the agricultural functions according to the differences between the three agricultural regional types. The results show the following: (1) The changes in the basic functions of agriculture, the intermediate functions of agriculture, and the advanced functions of agriculture were different from the perspectives of both time and space. (2) The carbon effects produced by the areas with weak agricultural functions, the areas with medium agricultural functions, and the areas with strong agricultural functions were different. (3) The evolution of agricultural production types aggravated the grain risk in the PRD, and urban agriculture has potential in improving food security. (4) Based on the regional types of agricultural functions and considering the constraints of land and water, strategic suggestions such as integrating natural resources, improving utilization efficiency, upgrading technical facilities, and avoiding production pollution are put forward. (5) The green and low-carbon transformation of urban agriculture has its boundaries. The positive effects of the factors, namely the innovation of agricultural production methods, the change in agricultural organization modes, the impact of market orientation, and the transfer of the agricultural labor force, is limited. The findings of this paper provide valuable and meaningful insights for academia, policy makers, producers, and ultimately for the local population in general, driving the development of urban agriculture in a low-carbon and sustainable direction.

**Keywords:** urban agricultural multifunctionality; agricultural carbon effect; SOFM network modeling; Granger causality analysis; the PRD

## 1. Introduction

With the rapid development of urbanization and industrialization, urban expansion and arable land loss have led to a fierce competition for land resources and food security issues between urban and rural areas [1]. The surge in urban populations and the improvement of residents' income levels have promoted a growth in total consumption, as well as the structural upgrading of agricultural products [2]. However, excessive reliance on natural resources and excessive investment in chemicals have led to environmental problems such as agricultural non-point source pollution and agricultural ecosystem degradation [3]. As an important paradigm for future agricultural development, urban agriculture is of great significance in leveraging agricultural comparative advantages, expanding space for

agricultural development, meeting consumer demand upgrades, and alleviating resource and environmental pressures [4,5]. Urban agriculture refers to the production activities related to agriculture in the metropolis or the urban space around the metropolitan area [6]. The difference between urban agriculture and traditional agriculture is that, as an organic part of the urban economy, society, and ecosystem, urban agriculture closely serves the city by relying on advantageous resources such as science and technology, talents, markets, information, capital, equipment, etc. The improvement of transport infrastructure and production technology makes urban agriculture more competitive [7]. In the process of urbanization, the alternating cycle between the socio-economic system and the ecological environment system has driven the transformation of agricultural production modes [8]. Agriculture has shifted from traditional planting to multifunctional, meeting the diverse demands of urban residents for products and services [9]. This restructuring has gone beyond traditional production models based on pure commodity production and has transformed into a new production system that provides consumers with other goods and services [10]. For the research on the multifunctional transformation of urban agriculture in China, the existing literature mainly focuses on the development stage of urban agriculture and the evolution of production characteristics [11], the spatial distribution of urban agricultural functions and its influencing factors [12], the synergy trade-off relationship between urban agricultural functions [13], and urban agricultural landscape ecology issues [14].

Agriculture attracts much attention due to its dual roles of carbon source and carbon sink in carbon neutralization-related research [15]. In human agricultural activities, on the one hand, the excessive use of agricultural inputs such as pesticides, fertilizers, and agricultural films, the wide application of agricultural machinery, soil ploughing, the processing and circulation of agricultural products, and the treatment and utilization of agricultural waste all lead to agricultural greenhouse gas emissions to varying degrees [16]. On the other hand, as a huge carbon pool, the plantation production system plays an important role in maintaining the agricultural function of soil as a carbon sink, and improving the soil's organic carbon level [17]. In terms of the research on the agricultural carbon effects in China, the existing literature mainly focuses on the distribution of and changes to agricultural carbon sources and sinks [18], the influencing factors of agricultural carbon emissions [19–21] and the spatial heterogeneity of different influencing factors [22], the relationship between agricultural carbon emissions and specific variables related to agricultural sustainable development such as space utilization efficiency [23] and agricultural ecological efficiency [24], and the impact of residents' dietary consumption structure [25] as well as agricultural production and operation methods [26,27] on the carbon footprint of the agri-food sector. Compared with traditional agricultural production, urban agriculture improves production efficiency with the help of scientific and technological advantages [28], diversified sales channels to increase the economic benefits [29], and the derived service industry and its complete industrial chain to produce positive social effects [30]. For urban agriculture, green and efficient production modes, a sustainable ecological economy, and gradually deepening low-carbon concepts all result in carbon effects different from traditional agriculture.

In terms of research content, the existing literature mainly discusses multifunctional transformation and the carbon effects of urban agriculture separately, but fails to link the two with each other, ignoring that the carbon effects of urban agriculture in the process of multifunctional transformation are different from traditional agriculture. As for the research on agricultural carbon effects, existing studies mainly focus on the quantitative assessments of the spatial distribution and influencing factors of carbon emissions, but they do not take into account that, as a regional system where humans and nature coexist in harmony, urban agriculture has dual attributes of natural reproduction and economic reproduction. Although the pursuit of productivity may lead to carbon emissions, the carbon sink function of crops and soils contributes to the carbon balance of the ecosystem. From the perspective of spatial and temporal continuity and heterogeneity, most studies mainly select time cross-section data, which makes it difficult to divide the continuous

process of the multifunctional transformation of urban agriculture into stages. Moreover, the existing literatures fail to consider the spatial heterogeneity of urban agricultural multifunctional regions. In other words, different types of urban agricultural regions may result in different carbon effects due to differences in production and management modes, natural resource endowments, social and economic development, etc.

Based on this, taking the Pearl River Delta as a typical case area, we constructed a multifunctional index system of urban agriculture, quantitatively evaluating the spatio-temporal evolution process of the urban agriculture function at the county level in the PRD for 19 consecutive years from 2002 to 2020. SOFM network modeling was adopted to divide the regional functional types of urban agriculture. Combined with the calculation of carbon emissions and the carbon sequestration of urban agriculture at the county level, a Granger causality test was applied to explore the differences in carbon effects of different urban agricultural regions. Finally, the carbon effects caused by agricultural functions were comprehensively analyzed in order to deepen the theory of agricultural multifunctionality and provide scientific support for the sustainable development of urban agriculture.

## 2. Materials and Methods

### 2.1. Study Area

The Pearl River Delta region is located in the middle of Guangdong Province (Figure 1), covering an area of 54,770 km$^2$. The climate in the PRD is mild and humid. This area is the subtropical monsoon climate zone, with a low and flat terrain [31], fertile soil, and a good water network with many branches [32], which is very suitable for agricultural development. The PRD is at the forefront of reform and opening up, and is the transportation hub and economic center of South China. Rapid economic development and urbanization have promoted the regional agricultural transformation, so that the transformation of agricultural functions and the evolution of the agro-ecological environment are highly typical and representative [33]. Urban agriculture in the PRD has a high level of modernization and socialized service [34]. Based on regional advantages and resource advantages, it is the future development direction for developing efficient product-type agriculture, ecological technology-type agriculture, and characteristic service-oriented agriculture [35]. The rapid growth of the population and the economic development have triggered the issues of food security and ecological environment in the PRD. While ensuring the total output value of agriculture, the region needs to consider the sustainability of agricultural production modes and avoid the huge ecological pressure caused by high-carbon operation modes. Urban agriculture provides a new path for the future of agricultural development in the PRD. It is of great practical significance to study the carbon effects of urban agriculture in the PRD region.

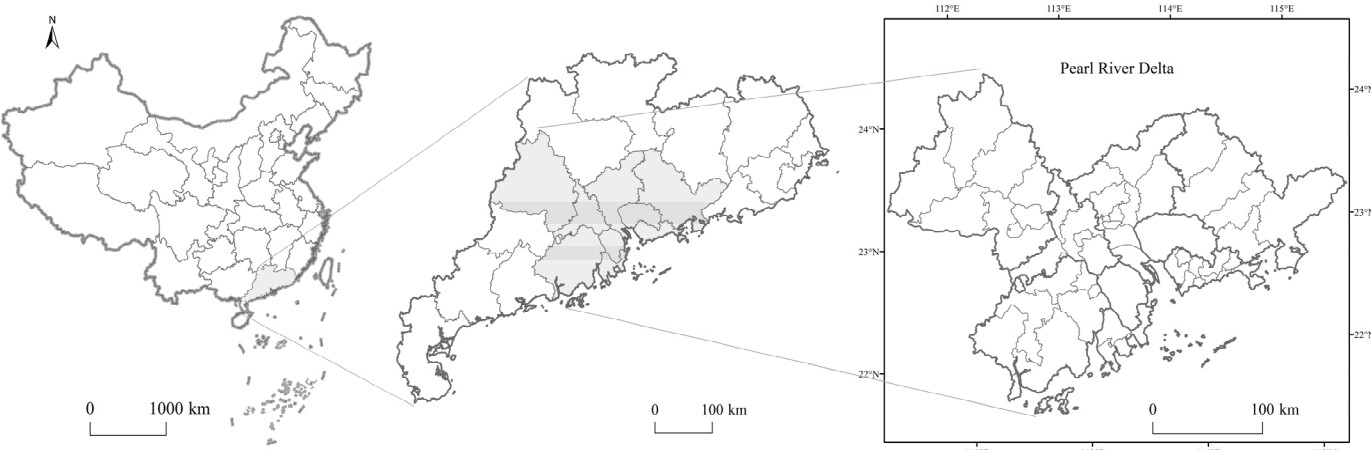

**Figure 1.** The location of the Pearl River Delta.

### 2.2. Data Sources

The data for the PRD from 2002 to 2020 are from the Guangdong Rural Statistical Yearbook, the Guangdong Statistical Yearbook, and statistical yearbooks by municipality (2003–2021). The NPP data for 2002–2020 are from MOD17A3HGF Version (https://lpdaac.usgs.gov/product_search/?view=listhttps://lpdaac.usgs.gov/product_search/?vie6.0w=list, accessed on 18 Match 2022). The 30 m annual land cover dataset in China (https://zenodo.org/record/4417810#.YShGWugzbBU, accessed on 18 Match 2022) was used to extract the data on farmlands. The administrative vector data are from the standard map with the approval number of GS(2019)1698 on the standard map service website of the Ministry of Natural Resources of China (https://www.resdc.cn/, accessed on 18 Match 2022).

### 2.3. Methods

#### 2.3.1. Construction of Multifunctional Index System for Urban Agriculture

Urban agriculture has three major functions: production, economy, and society [36–38]. Among them, the production function of urban agriculture is reflected in the increasing food supply capacity, providing fresh agricultural products and seasonal vegetables and fruits to ensure the safety of both food quantity and quality [39], which is the most basic function of urban agriculture. In terms of economic functions, by promoting the upgrading of industrial structure, urban agriculture improves the output and value of agricultural products and promotes the improvement of labor productivity and cultivated land productivity [40]. In terms of social functions, on the one hand, by utilizing a variety of agricultural resources, urban agriculture extends the agricultural industry chain, improves the level of the agricultural service industry, and expands the channels for increasing farmers' income. On the other hand, by retaining the production and farming characteristics of traditional agriculture, urban agriculture provides agriculture-related employment opportunities for migrant workers and surplus labor from other industries, and promotes the integration of groups with low labor skills into the city [5]. Accordingly, we combined the actual development of urban agriculture in the PRD to construct an urban agricultural function evaluation index system consisting of three functional indicators: production, economy, and society (Table 1).

**Table 1.** The index system of urban agricultural functions.

| Function | Index | Calculation formula | Weight |
|---|---|---|---|
| Production Function | Cultivation index | Area of cultivated land/Land area | 10.11% |
| | Grain crop output per unit area | Yield of grain crops/Sown area of grain crops | 3.40% |
| | Per capita share of grain crops | Yield of grain crops/Permanent population at year-end | 34.36% |
| | Per capita share of fruits and vegetables | (Gross output of fruits + Gross output of vegetables)/Permanent population at year-end | 21.28% |
| | Per capita share of agricultural products in animal husbandry | (Output of milk + Output of poultry eggs + Output of meat + Output of honey)/Permanent population at year-end | 30.84% |
| Economic Function | Agricultural output value per capita | Gross output value of agriculture, forestry, animal husbandry, and fishery/Permanent population at year-end | 14.11% |
| | Proportion of gross output value of agriculture, forestry, animal husbandry, and fishery | Gross output value of agriculture, forestry, animal husbandry and fishery/Gross domestic product | 19.56% |
| | Cultivated land productivity | Gross output value of agriculture, forestry, animal husbandry, and fishery/Area of cultivated land | 48.19% |

**Table 1.** *Cont.*

| Function | Index | Calculation formula | Weight |
|---|---|---|---|
| | Agricultural labor productivity | Gross output value of agriculture, forestry, animal husbandry, and fishery/Total number of employed persons at year-end | 18.15% |
| Social Function | Per capita income level of rural residents | Per capita annual disposable income of rural residents | 26.50% |
| | Employment structure level | Labor force in the primary industry/Rural labor force | 16.18% |
| | Agricultural service level | Proportion of service industry for agriculture in gross output value of agriculture, forestry, animal husbandry, and fishery | 57.32% |

To eliminate the influence of dimension, nature difference, and the order of magnitude among indicators, the range standardization method was adopted. The calculation formula is as follows:

$$x_{ij} = (X_{ij} - X_{jmin})/(X_{jmax} - X_{jmin}) \tag{1}$$

where $X_{ij}$, $X_{jmin}$, $X_{jmax}$ and $x_{ij}$ are the original value, minimum value, maximum value and standardized value of the j-th index in the i-th area, respectively.

The entropy weight method is used to determine the index weight [41]. The calculation formula is as follows:

$$E_j = -(\ln(n))^{-1} \sum_{i=1}^{n} P_{ij} \ln(P_{ij}) \tag{2}$$

where $E_j$ is the index entropy of j, which is used to measure the amount of effective information provided by the data. The larger the entropy value, the greater the degree of disorder, and the smaller the amount of effective information (that is, the smaller the weight), and vice versa. n is the total number of objects.

$$P_{ij} = x_{ij} / \sum_{i=1}^{n} x_{ij} \tag{3}$$

where $P_{ij}$ is the proportion of the i-th sample indicator in the j-th index.

$$w_j = (1 - E_j) / \sum_{j=1}^{m} (1 - E_j) \tag{4}$$

where $w_j$ is the weight of the j-th index, and m is the number of functional indicators.

The calculation formula of each function score of each administrative region is as follows:

$$s_i = \sum_{j=1}^{m} x_{ij} w_j \tag{5}$$

where $s_i$ is the score of each function.

2.3.2. Estimation Method of Carbon Emissions and Carbon Sequestration

The main sources of carbon emissions from agricultural inputs are pesticides, agricultural film, chemical fertilizers, agricultural machinery, agricultural irrigation, and farmland tillage. The formula for estimating agricultural carbon emissions is as follows:

$$E = \sum E_i = \sum (T_i \times Q_i) \tag{6}$$

where E represents the total carbon emissions from agriculture; $E_i$ represents the carbon emissions of the i-th carbon source; $T_i$ represents the amount of the i-th carbon source; and $Q_i$ represents the carbon emission coefficient of the i-th carbon source (Table 2).

**Table 2.** Carbon emission coefficient.

| Carbon Source | Carbon Emission Coefficient |
|---|---|
| Agricultural pesticides | 4.9341 kg(C)·kg$^{-1}$ |
| Plastic film in agriculture | 5.1800 kg (C)·kg$^{-1}$ |
| Chemical fertilizers | 0.8956 kg(C)·kg$^{-1}$ |
| Agricultural irrigation | 266.4800 kg(C)·hm$^{-2}$ |
| Farmland tillage | 312.6000 kg(C)·hm$^{-2}$ |
| Diesel oil in agriculture | 0.5927 kg(C)·kg$^{-1}$ |
| Agricultural ploughing | 16.4700 kg(C)·hm$^{-2}$ |
| Agricultural electricity conversion | 0.1800 kg(C)·kw$^{-1}$ |

Note: These data are from the carbon emission coefficient released by the IPCC [42].

Net primary productivity (NPP) is the accumulated organic dry matter yield of green plants per unit of time and area after subtracting autotrophic respiration [43]. NPP is the main factor to determine the carbon sink of the ecosystem and regulate the ecological process [44]. We estimated the carbon sequestration based on the NPP, and the NPP is estimated through the light use efficiency model, the Carnegie–Ames–Stanford approach (CASA) model [45]. The basic principles of the model are as follows:

$$NPP(x,t) = APAR(x,t) \times \varepsilon(x,t) \tag{7}$$

where $NPP(x, t)$ is the net initial productivity of pixel x at time t (gC/m$^2$), APAR is photosynthetically active radiation absorbed by plants (MJ/m$^2$), $\varepsilon$ is the light efficiency, t is the time, and x is the spatial location.

$$APAR(x,t) = SOL(x,t) \times FPAR(x,t) \times 0.5 \tag{8}$$

where $SOL(x, t)$ is the total solar radiation of pixel x in time t. *FPAR* is the absorption ratio of incident photosynthetically active radiation (PAR) by vegetation layer. The constant term 0.5 represents the ratio of solar effective radiation (wavelength 0.38–0.71) that vegetation can use to total the solar radiation. There is a linear relationship between FPAR and NDVI within a certain range, and the corresponding FPAR can be obtained according to NDVI [46].

The carbon sequestration is calculated using the following formula:

$$\text{NPP}' = (\text{NPP}/0.5) \times 1.62 \tag{9}$$

Each gram of dry matter can fix 1.62 g $CO_2$. The dry matter content accounts for about 45–55% of the NPP content, and the average value of 50% is selected in this study.

The carbon sequestration of cultivated land can be obtained by adding the NPP corresponding to the cultivated land in land-use data using the ArcGIS grid calculator. According to the natural breakpoint method, the urban agricultural functions and carbon effects can be divided into five levels, namely low-value area, medium–low-value area, median area, medium–high-value area, and high-value area in ascending order.

2.3.3. Self-Organizing Feature Maps Network Modeling

Self-organizing feature maps (SOFM) network modeling was used to divide the urban agricultural functional region of the PRD. SOFM has the characteristics of topological structure maintenance, self-organizing probability distribution, non-supervised learning and visualization, strong fault tolerance, etc. SOFM has been widely used in the research division and classification in disciplines such as geography and land science [47]. The calculation steps are as follows. First, initialize the weights and assign random decimals to all connection weights from the input node to the output node. Second, define a new network input mode and randomly select an input sample from the sample set. Next, calculate the Euclidean distance between the input sample and each output neuron, and

select the output neuron with the maximum similarity measure as the winning output unit. Then, modify the connection weights between the selected neuron and adjacent neurons. Finally, input new samples and repeat the learning process until a meaningful mapping is formed [48,49].

### 2.3.4. Granger Causality Test

The Granger causality test was used to investigate the relationship between carbon effects and urban agricultural multifunctionality. Granger (1969) proposed the definition of Granger causality based on the time order of events. If the prediction result of the occurrence of Y event on the occurrence of X event is better than the prediction result of the occurrence of X event without prior conditions, then Y event is the Granger cause of X event [50]. Two time series $\{x_t\}$ and $\{y_t\}$ are set to compare whether the conditional expectation of X under the original information is different from that under the added information of Y; that is, in the following formula:

$$x_t = \sum_{i=1}^{\infty} \alpha_i x_{t-i} + \sum_{i=1}^{\infty} \beta_i y_{t-i} + \varepsilon_i \tag{10}$$

if there is at least one $i_0$, which makes $\beta_{i0} \neq 0$, then the variable y is the Granger cause of x.

Sims (1980) proposed a VAR model guided by generational optimization. The instability of variables in the VAR model may lead to the instability of the estimator. Therefore, it is necessary to pay attention to the stationarity of time series variables when applying the VAR model [51]. In this paper, a unit root test of ADF is performed on the variables to determine its stability. Then the residual-based EG method is used to determine the cointegration relationship of the stationary series.

Regarding the stability of the model as a whole, Hamilton (1994) pointed out that if all the adjoint matrix vectors are strictly less than 1, then the VAR model is stable [52]. In this paper, the multi-criteria joint determination method is used to determine the lag order. LR, FPE, AIC, SC, HQ criteria and AR Roots Graphs are used to test whether the VAR model constructed between the two variables is stable and effective. Only when all the unit roots are within the unit circle, the constructed VAR model is stable and effective. After the stable VAR model is built, the Granger causality test is carried out. If the P value is less than 0.05, there is a Granger causality between the two variables.

Stock and Watson (2009) conducted dynamic factor modelling [53]. To more intuitively analyze the impact of one endogenous variable on other endogenous variables, the impulse response function is used to describe the dynamic interaction between variables in the short term after Granger causality analysis.

## 3. Results

### *3.1. The Multifunctional Transformation Process of Urban Agriculture*

The production function of urban agriculture in the PRD experienced a stepwise decline process. From 2002 to 2006, it fluctuated between 0.219 and 0.229, with an average value of 0.223; from 2007 to 2016, it stabilized between 0.185 and 0.190, with an average value of 0.188; from 2017 to 2020, it dropped from 0.176 to 0.170, with an average value of 0.173. The economic function experienced a slow and fluctuating increase. From 2002 to 2005, it increased from 0.064 to 0.079, with an average value of 0.071; from 2006 to 2014, except for 2008, which was 0.092, in the rest of the years it fluctuated between 0.069 and 0.076, with an average value of 0.073; from 2015 to 2020, it increased from 0.082 to 0.094, with an average value of 0.085. The social function significantly increased from 0.101 to 0.189 between 2002 and 2020, surpassing the declining production function in 2018, with an average value of 0.139 over 19 consecutive years (Figure 2).

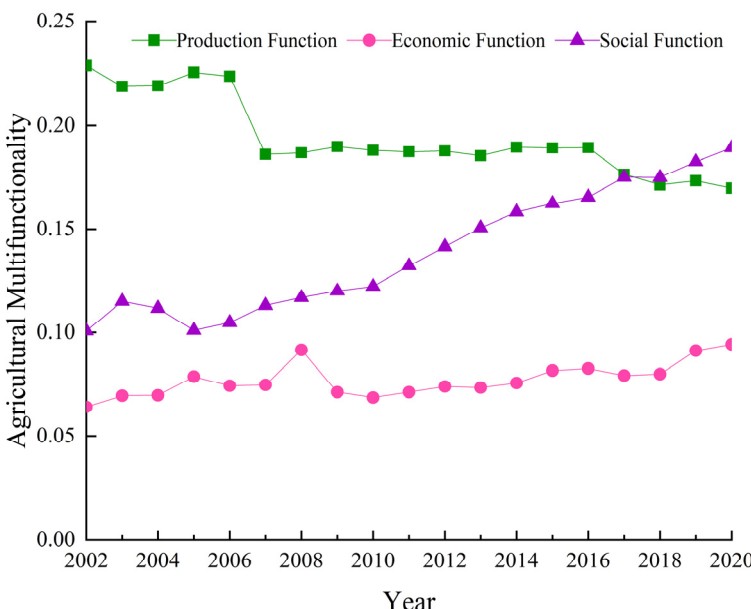

**Figure 2.** Urban agricultural multifunctionality in PRD from 2002 to 2020.

The regions with weak production functions of urban agriculture in the PRD expanded from the inside out, and the spatial heterogeneity of the "center—periphery" gradually became obvious, which means that the gap in the agricultural production function between the core area and the peripheral area was widening (Figure 3). From 2002 to 2006, the regions with weak production functions were limited to Shenzhen, Dongguan, the central urban areas of Guangzhou, the east of Foshan, Pengjiang District of Jiangmen, Xiangzhou District, and Jinwan District of Zhuhai, while other areas had a strong agricultural production function. From 2007 to 2016, the regions with weak production functions expanded to include Shenzhen, Dongguan, the center and south of Guangzhou, the east of Foshan, Pengjiang District and Jianghai District of Jiangmen, Zhongshan, Xiangzhou District, and Jinwan District of Zhuhai. From 2017 to 2020, the agricultural production function in the core area of the PRD further weakened, while that in the peripheral areas further strengthened. The spatial heterogeneity characteristics of the "center—periphery" gradually strengthened.

The region with strong economic functions of urban agriculture in the PRD slowly spread from the outside in (Figure 4). From 2002 to 2005, the region with weak economic functions included Shenzhen, Dongguan, Huicheng District and Huiyang District of Huizhou, the center and south of Guangzhou, the east of Foshan, Pengjiang District and Jianghai District of Jiangmen, Zhongshan, and Zhuhai. The economic function of Zhaoqing in the peripheral areas of the PRD completed the transition from the median area to the medium–high-value area. From 2006 to 2014, in the peripheral areas of the PRD, Fengkai County of Zhaoqing steadily transformed into the high-value area, and Taishan City of Jiangmen and Doumen District of Zhuhai transformed from the median area to the medium–high-value area. From 2015 to 2020, the region with strong economic functions spread to include the adjacent regional units of Fengkai County, Taishan City, and Doumen District. The economic function in Zhaoqing gradually changed from the medium–high-value area to the high-value area. The economic function in Enping City and Kaiping City of Jiangmen changed from the median area to the medium–high-value area. The economic function in Xiangzhou District of Zhuhai, Shunde District of Foshan, and Longmen County of Huizhou also changed to the medium–high-value area.

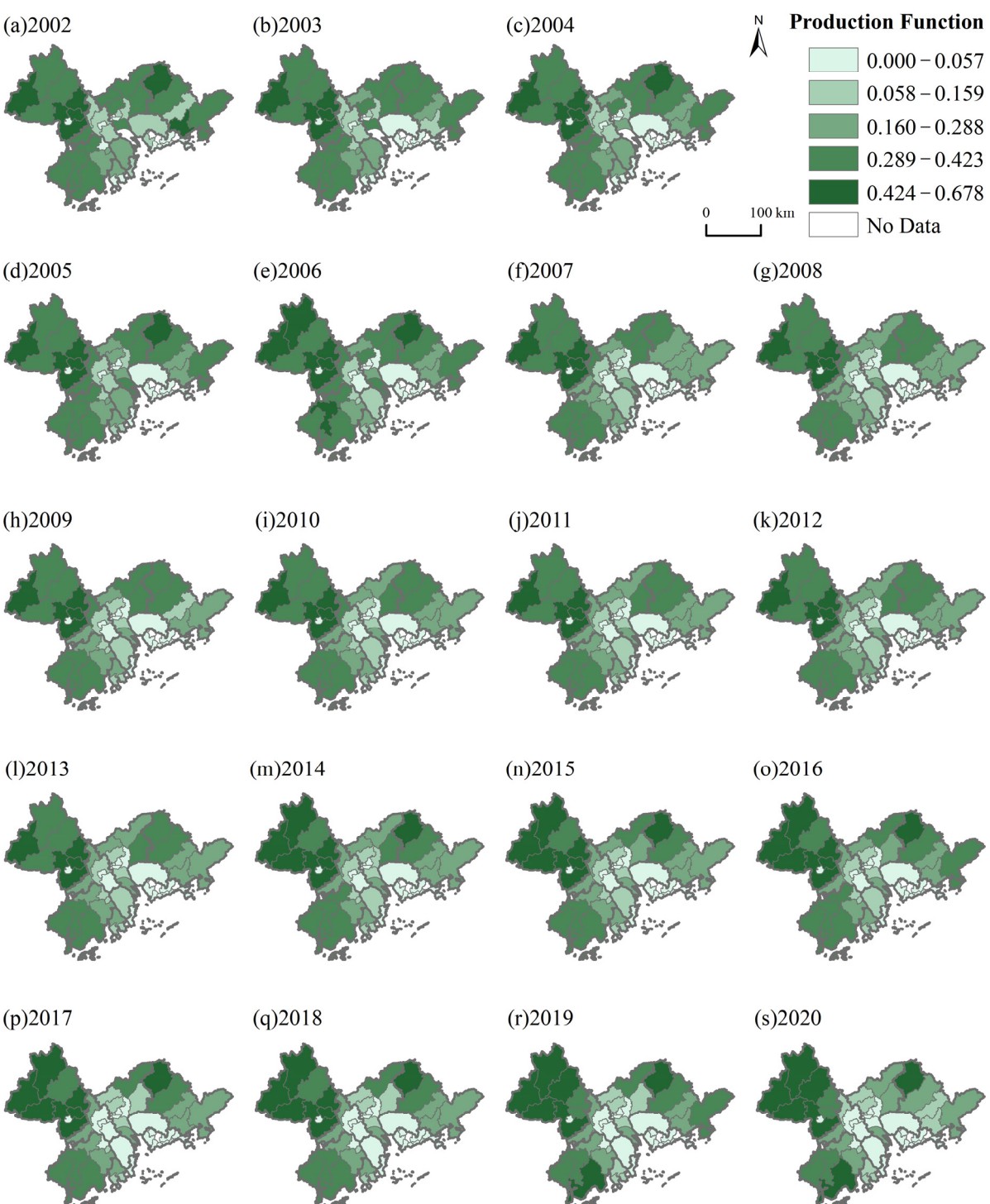

**Figure 3.** The production function of urban agriculture in PRD from 2002 to 2020.

The region with strong social functions of urban agriculture in the PRD expanded from the inside out (Figure 5). From 2002 to 2012, the region with strong social functions expanded outward from the center and north of Guangzhou, and the social function of the PRD basically realized the transition from the medium–low-value area to the median area. From 2013 to 2020, the social function was further enhanced. The region with strong social functions expanded from the center of Guangzhou and the north of Zhaoqing, and the social function of the PRD basically realized the transition from the median area to the medium–high-value area.

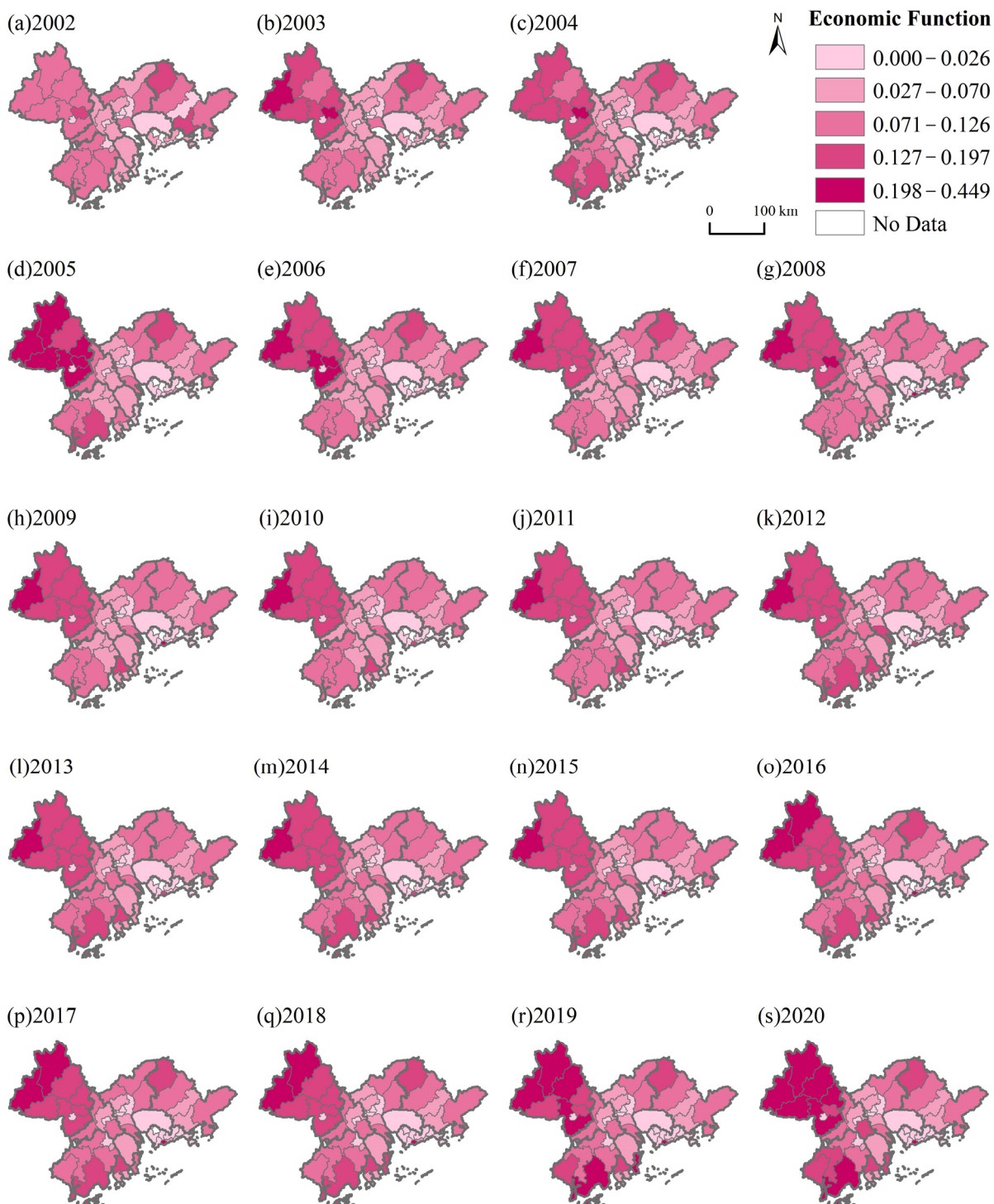

**Figure 4.** The economic function of urban agriculture in PRD from 2002 to 2020.

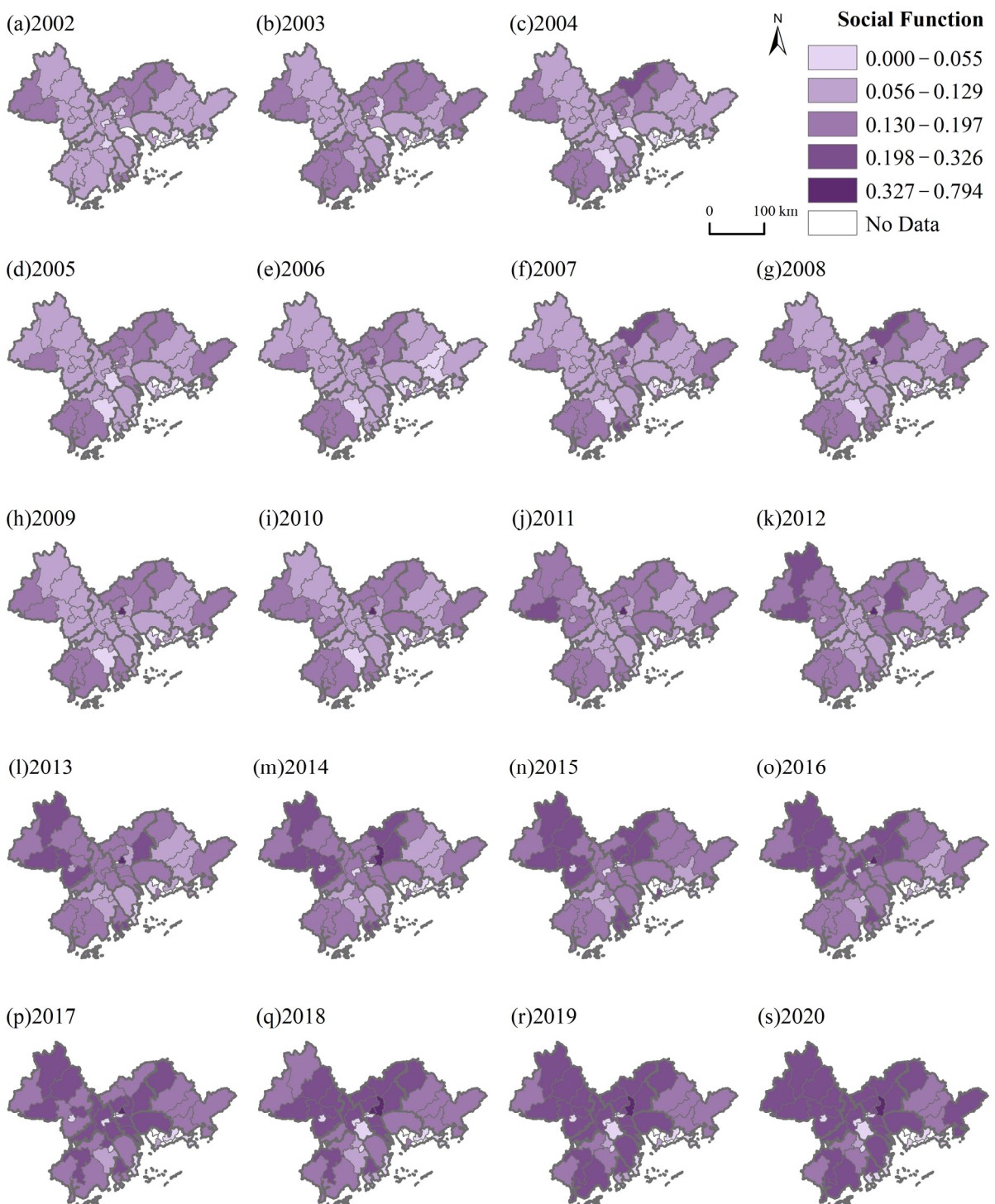

**Figure 5.** The social function of urban agriculture in PRD from 2002 to 2020.

### 3.2. The Carbon Effect Evolution Process of Urban Agriculture

Urban agriculture in the PRD is a huge carbon sink as a whole (Figure 6). From 2002 to 2020, the average carbon sequestration was $8.6 \times 10^6$ t, the average carbon emissions was $1.1 \times 10^6$ t, and the carbon sequestration was about 7.8 times the carbon emission. From 2002 to 2013, the carbon sequestration experienced a fluctuating decrease from $9.3 \times 10^6$ t to $7.8 \times 10^6$ t, with an average annual change rate of 1.5%, and then steadily increased to $8.9 \times 10^6$ t in 2020, with an average annual change rate of 2.0%. From 2002 to 2010, carbon emissions increased slowly from $1.1 \times 10^6$ t to $1.2 \times 10^6$ t, with an average annual change

rate of 1.1%, and then dropped steadily to $1.0 \times 10^6$ t in 2020, with an average annual change rate of 1.7%.

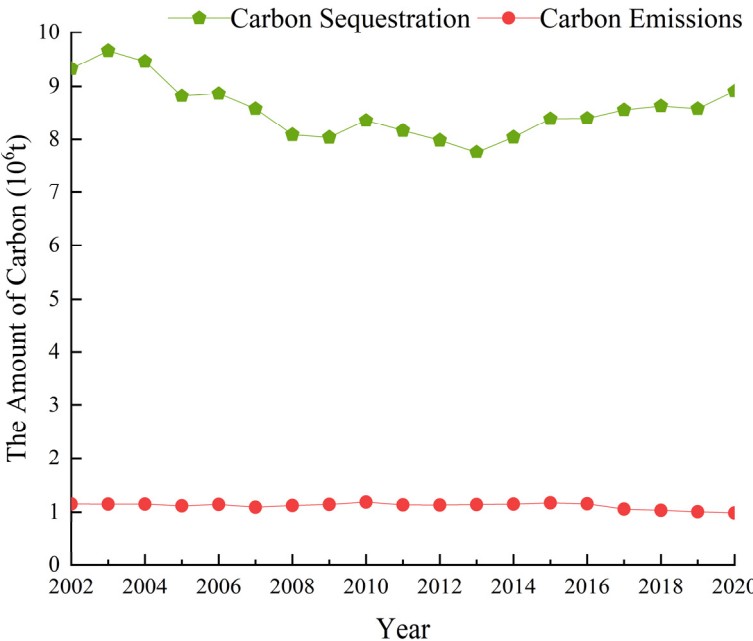

**Figure 6.** Urban agricultural carbon effects in PRD from 2002 to 2020.

The agricultural carbon emissions in the PRD were mainly concentrated in the peripheral areas, and the range continued to shrink. The amount of agricultural carbon emissions in the peripheral areas first increased and then decreased, while that in the central areas continued to decline (Figure 7). From 2002 to 2004, the region with weak agricultural carbon emissions only included Shenzhen, the central urban areas of Guangzhou (Liwan District, Haizhu District, Tianhe District and Huangpu District), the central urban areas of Foshan (Chancheng District), the central urban areas of Zhaoqing (Duanzhou District and Dinghu District), the central urban areas of Zhuhai (Xiangzhou District and Jinwan District), and the central urban areas of Jiangmen (Jianghai District and Pengjiang District). The amount of agricultural carbon emissions in other regions was large. From 2005 to 2013, apart from the above-mentioned central urban areas, agricultural carbon emissions in Huiyang District of Huizhou, Panyu District of Guangzhou, and Shunde District of Foshan also gradually fell to the medium–low-value area. Meanwhile, the amount of agricultural carbon emissions in Dongguan, Zhongshan, Nanhai District of Foshan, and Xinhui District of Jiangmen also decreased, while that of Zhaoqing outside the PRD increased. From 2014 to 2020, the agricultural carbon emissions in Dongguan, Foshan, and Zhuhai have gradually decreased to below the medium–low-value area. The agricultural carbon emissions of Huizhou, Zhaoqing, and Jiangmen outside the PRD also declined, and the concentration range of agricultural carbon emissions further shrank.

The agricultural carbon sequestration in the PRD was mainly concentrated in the peripheral areas, and the carbon sink-intensive areas first contracted significantly and then expanded slightly (Figure 8). From 2002 to 2006, the concentration range of agricultural carbon sequestration was Huizhou, Dongguan, Zengcheng District and Conghua District of Guangzhou, Huaiji County, Fengkai County, Gaoyao District and Sihui City of Zhaoqing, Jiangmen, and Zhongshan. From 2007 to 2013, carbon sink-intensive areas mainly shrank in Longmen County of Huizhou, Dongguan, Sihui of Zhaoqing, Heshan City and Xinhui District of Jiangmen, and Zhongshan. From 2014 to 2020, agricultural carbon sinks only recovered slightly in Longmen County of Huizhou and Xinhui District of Jiangmen.

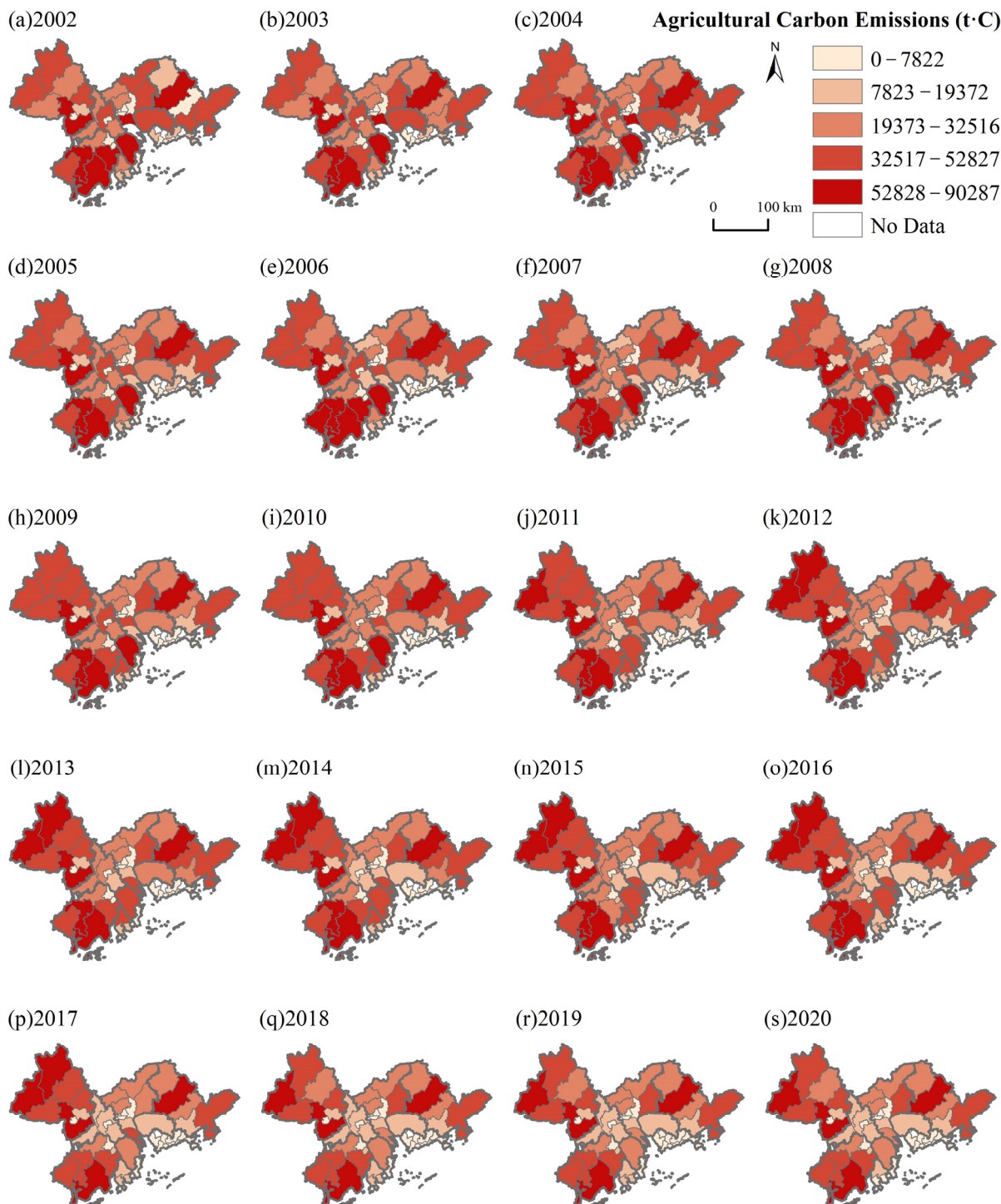

**Figure 7.** Carbon emissions of urban agriculture in PRD from 2002 to 2020.

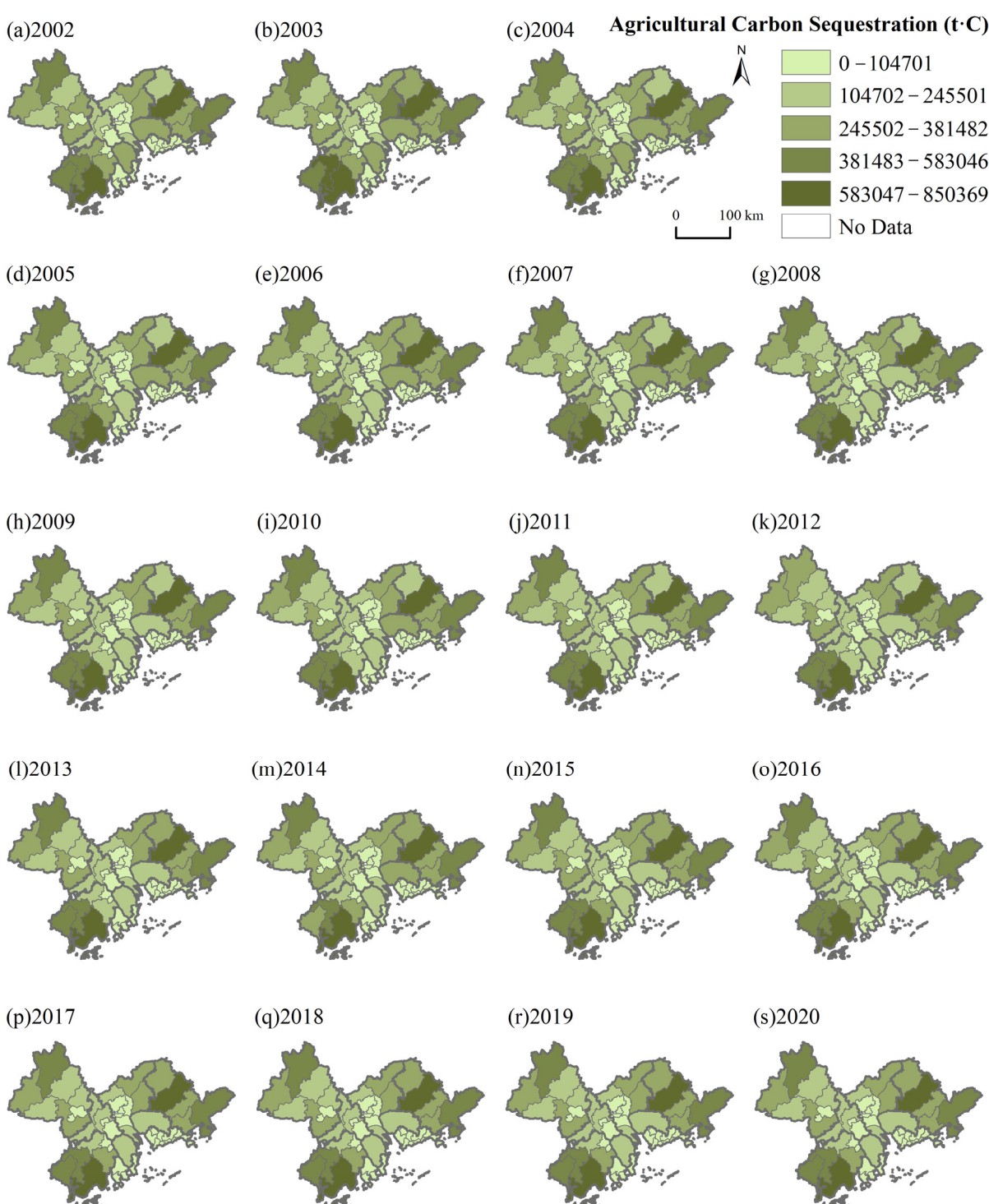

**Figure 8.** Carbon sequestration of urban agriculture in PRD from 2002 to 2020.

*3.3. Classification of Urban Agricultural Functional Regions and The Causal Test*

　　According to the three major functions of urban agriculture from 2002 to 2020, the PRD can be divided into areas with weak agricultural functions, areas with medium agricultural functions, and areas with strong agricultural functions from the center to the periphery by using SOFM (Figure 9). Granger causality analysis and the impulse response function were used to test the causal relationship between agricultural functions and agricultural carbon effects in the three types of agricultural regions, so as to clarify the differences in carbon effects produced in different agricultural regions.

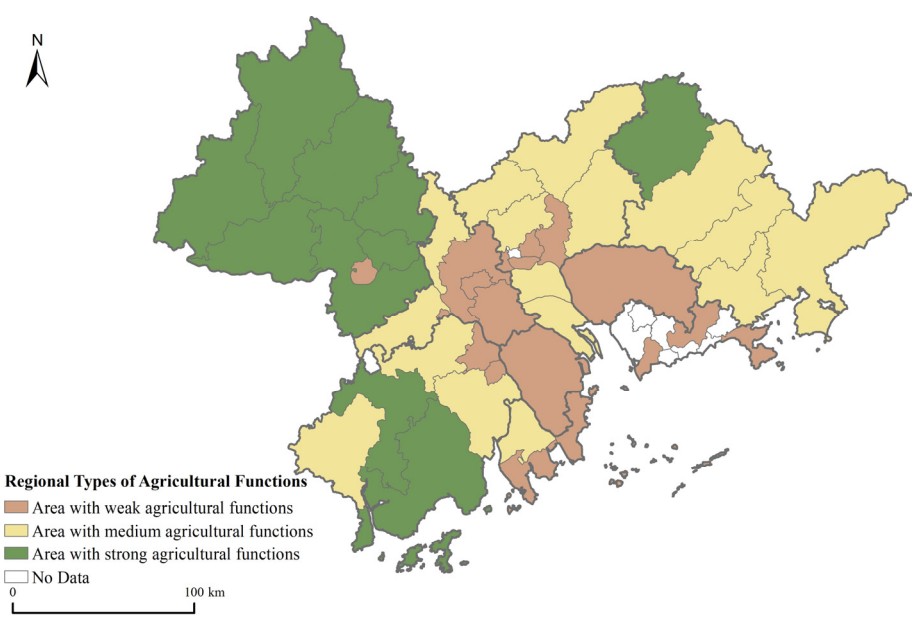

**Figure 9.** Regional types of urban agricultural functions in PRD.

The Granger causality test was carried out on the agricultural functions and carbon effects in the areas with weak agricultural functions (Table 3), and the impulse response functions were drawn (Figure 10). The results are as follows. The production function is not the Granger cause of carbon emissions. The amount of carbon emissions has the largest response to the economic function (+0.063) when the lag period is four, and has the largest response to the social function (−0.037) when the lag period is two. The response of carbon sequestration to the production function is the largest when the lag period is two, which is −0.024. The response to economic function is the largest when the lag period is three, which is −0.013. The response to social function is the largest when the lag period is two, which is −0.026. In summary, the production function of the areas with weak agricultural functions has the effect of decreasing sinks, the economic function has the effects of increasing emission and decreasing sinks, and the social function has the effects of reducing emissions and decreasing sinks.

**Table 3.** Results of Granger causality test in the areas with weak agricultural functions.

| Variable | Lag Order for VAR Model | $p$ Value for Granger Causality Test |
|---|---|---|
| Production function → Carbon emissions | 1 | 0.7509 |
| Production function → Carbon sequestration | 1 | <0.0001 |
| Economic function → Carbon emissions | 3 | <0.0001 |
| Economic function → Carbon sequestration | 2 | 0.0053 |
| Social function → Carbon emissions | 1 | <0.0001 |
| Social function → Carbon sequestration | 1 | <0.0001 |

The area with weak agricultural functions located at the core of the PRD has the serious problem of cultivated land fragmentation. Therefore, its economic function has the carbon emission increase effect in spite of the advantages in technology and capital. From the perspective of scale effects, the fragmentation of cultivated land not only makes the scale and shape of the cultivated land unfavorable to agricultural machinery operations, increasing the carbon emissions caused by energy consumption, but also hinders the sharing of facilities and equipment among agricultural production and management entities, increasing the carbon emissions caused by resource waste. From the perspective of agricultural management, the fragmentation of cultivated land makes the spatial distribution of farmland complicated. It is difficult to realize not only the real-time monitoring and analy-

sis of data on farmland environment, crop growth, and resource consumption, but also the accurate control and management of elements such as water, fertilizers, and pesticides.

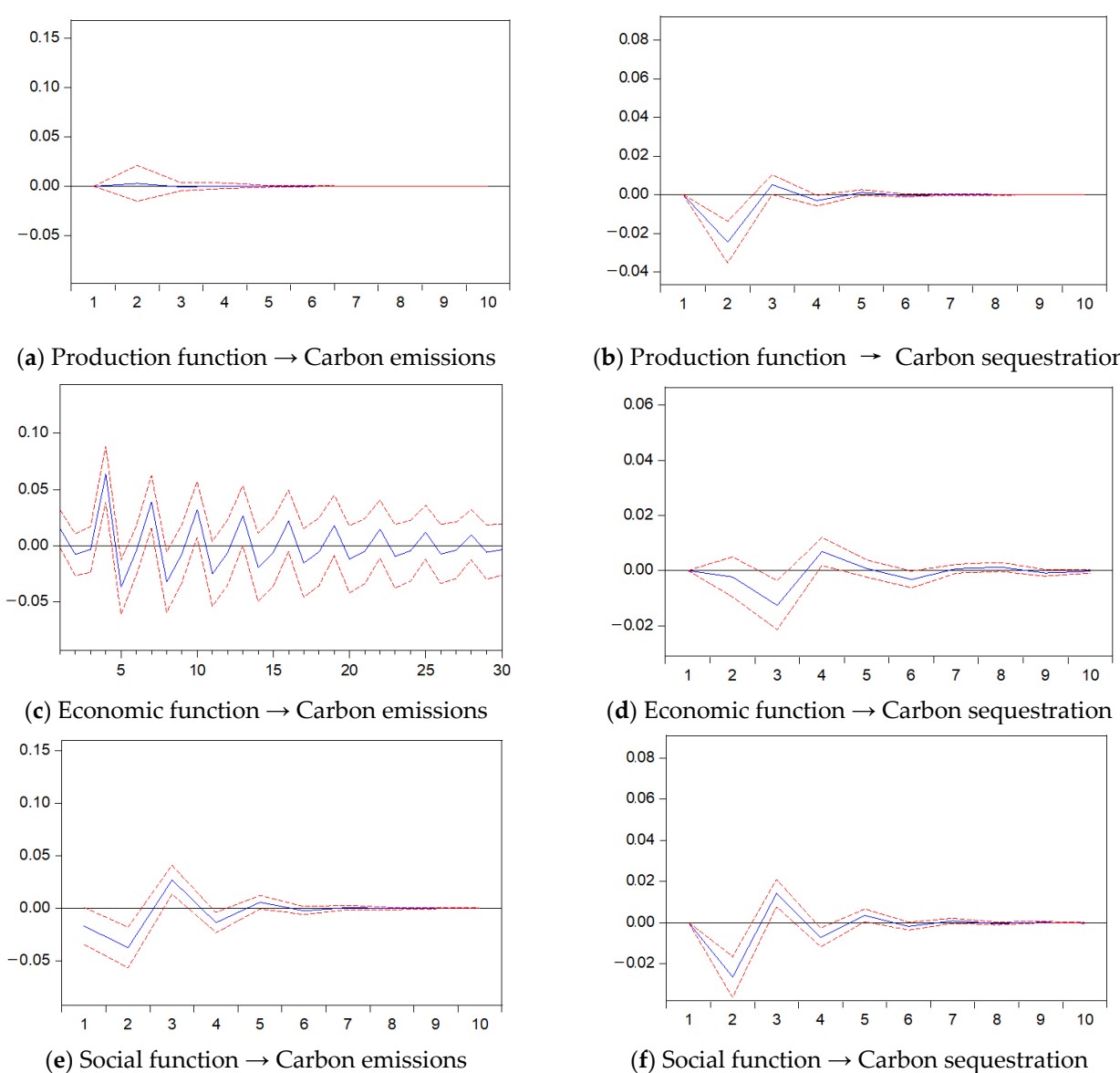

(**a**) Production function → Carbon emissions

(**b**) Production function → Carbon sequestration

(**c**) Economic function → Carbon emissions

(**d**) Economic function → Carbon sequestration

(**e**) Social function → Carbon emissions

(**f**) Social function → Carbon sequestration

**Figure 10.** Impulse response functions of carbon effects and agricultural functions in the areas with weak agricultural functions.

The Granger causality test was carried out on agricultural functions and carbon effects in the areas with medium agricultural functions (Table 4), and the impulse response functions were drawn (Figure 11). The results show that the responses of carbon emissions to the production function, economic function and social function are the largest when the lag period is two, which are −0.041, −0.034 and −0.021, respectively. The responses of carbon sequestration to the production function, economic function, and social function also reach the maximum when the lag period is two, which are −0.060, −0.038, and −0.023, respectively. In summary, all three agricultural functions in the areas with medium agricultural functions have the carbon effects of emission reduction and sink reduction.

**Table 4.** Results of Granger causality test in the areas with medium agricultural functions.

| Variable | Lag Order for VAR Model | $p$ Value for Granger Causality Test |
|---|---|---|
| Production function → Carbon emissions | 1 | <0.0001 |
| Production function → Carbon sequestration | 1 | <0.0001 |
| Economic function → Carbon emissions | 1 | 0.0006 |
| Economic function → Carbon sequestration | 1 | 0.0003 |
| Social function → Carbon emissions | 2 | 0.0147 |
| Social function → Carbon sequestration | 2 | 0.0115 |

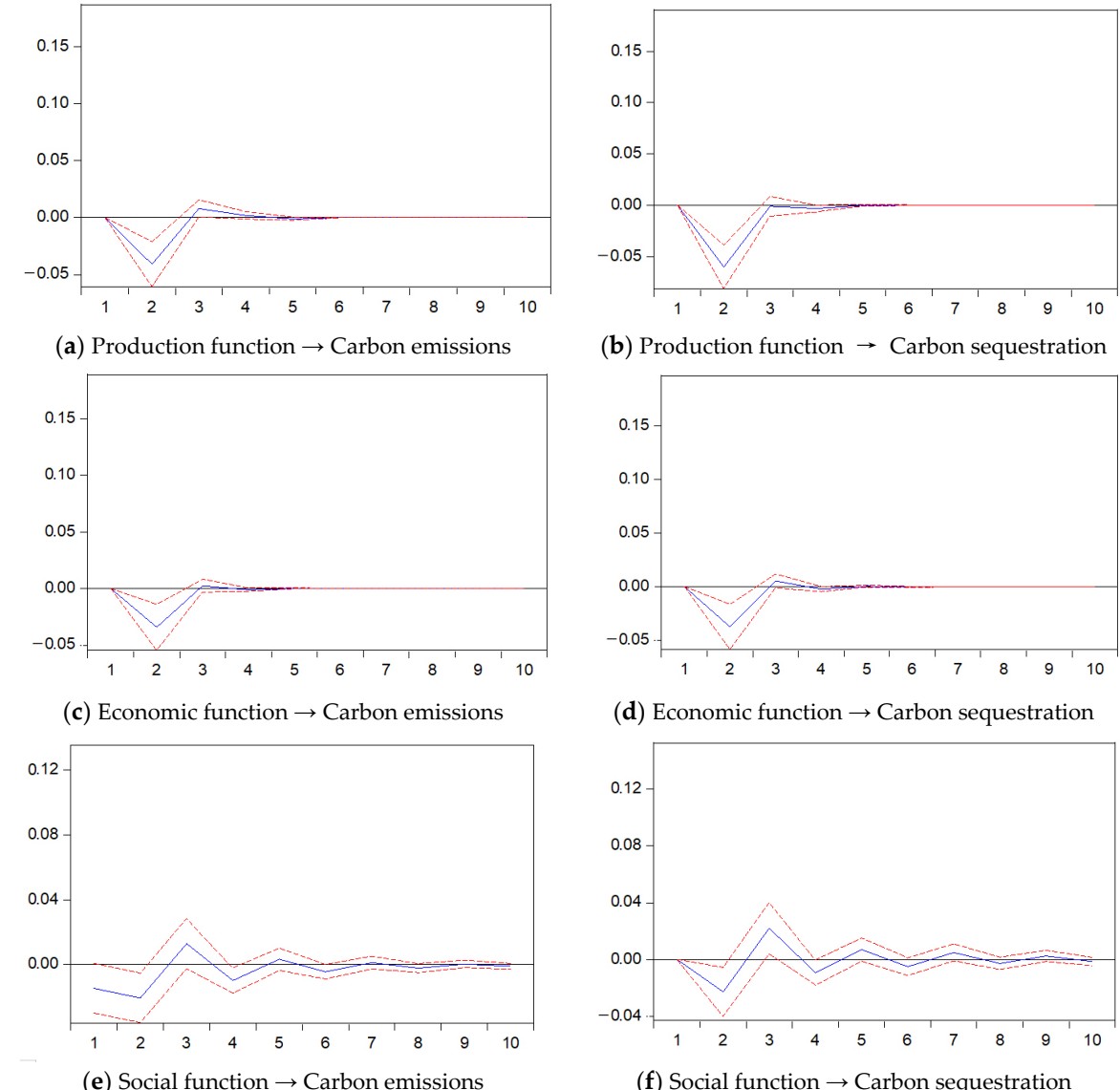

(**a**) Production function → Carbon emissions

(**b**) Production function → Carbon sequestration

(**c**) Economic function → Carbon emissions

(**d**) Economic function → Carbon sequestration

(**e**) Social function → Carbon emissions

(**f**) Social function → Carbon sequestration

**Figure 11.** Impulse response functions of carbon effects and agricultural functions in the areas with medium agricultural functions.

The areas with medium agricultural functions have a better foundation for agricultural modernization development. Advanced agricultural production modes, efficient agricultural management, and rational utilization of agricultural resources have jointly led to the carbon emission reduction effect of the production function in the areas with medium agricultural functions. From the perspective of agricultural production modes, agricultural production efficiency can be improved, and at the same time carbon emissions can be reduced in the areas with medium agricultural functions. The revolution of agricul-

tural science and technology is comprehensively promoted, and the innovative R&D and integrated application of digital technology and carbon reduction technology are strengthened, promoting the transformation of agricultural development's quality, efficiency, and impetus [54]. From the perspective of agricultural management, fine management can be achieved on the basis of ensuring the quality of grain production in the areas with medium agricultural functions. The agricultural production structure is optimized to develop new agricultural production and operation modes such as green agriculture, circular agriculture, and low-carbon agriculture [55]. From the perspective of agricultural resource utilization, the rational allocation, low carbon, and efficient utilization of agricultural resources can be realized in the areas with medium agricultural functions. The energy structure of agricultural production is optimized, and the efficiency and sustainable utilization of resources are improved, which are conductive to the agricultural ecological environment [56].

The areas with medium agricultural functions not only have relatively superior cultivated land resources and agricultural conditions, but also relatively developed economic foundations, resulting in the carbon emission reduction effect of the economic function. On the one hand, through the integration of modern information technology with the whole elements of agriculture, the whole industry chain, and the whole value chain [54], digital technology can be fully used and human wisdom can be integrated to participate in the decision making and control of the whole process of agricultural production [57], so that the areas with medium agricultural functions play the role of reducing emissions while realizing economic benefits. On the other hand, based on the decisive role played by the government's top-level design and institutional guarantees of the rational allocation of resources in the market, the areas with medium agricultural functions incorporate agricultural carbon emission reduction into the agricultural carbon trading market to promote its deep integration with regional resource endowments [58]. The relationship between the value of spatial ecological resources and the conversion of economic and monetary resources gets high attention [59], and ways and methods of financial service for the green and low-carbon development of agriculture are innovated [56].

The Granger causality test was carried out on agricultural functions and carbon effects in the areas with strong agricultural functions (Table 5), and the impulse response functions were drawn (Figure 12). The results show that the response of carbon emissions to the production function, economic function and social function reach the maximum when the lag period is two, which are +0.049, +0.047, and −0.018, respectively. The response of carbon sequestration to production function is the largest when the lag period is two, which is −0.047. The response to economic function is the largest when the lag period is three, which is −0.048. The response to social function is the largest when the lag period is two, which is −0.027. In summary, the production function and economic function of the areas with strong agricultural functions have the carbon effects of increasing emissions and decreasing sinks, and the social function has the carbon effects of reducing emissions and decreasing sinks.

**Table 5.** Results of Granger causality test in the areas with strong agricultural functions.

| Variable | Lag Order for VAR Model | $p$ Value for Granger Causality Test |
|---|---|---|
| Production function → Carbon emissions | 1 | 0.0005 |
| Production function → Carbon sequestration | 2 | <0.0001 |
| Economic function → Carbon emissions | 1 | 0.0004 |
| Economic function → Carbon sequestration | 2 | 0.0001 |
| Social function → Carbon emissions | 1 | 0.0346 |
| Social function → Carbon sequestration | 1 | 0.0033 |

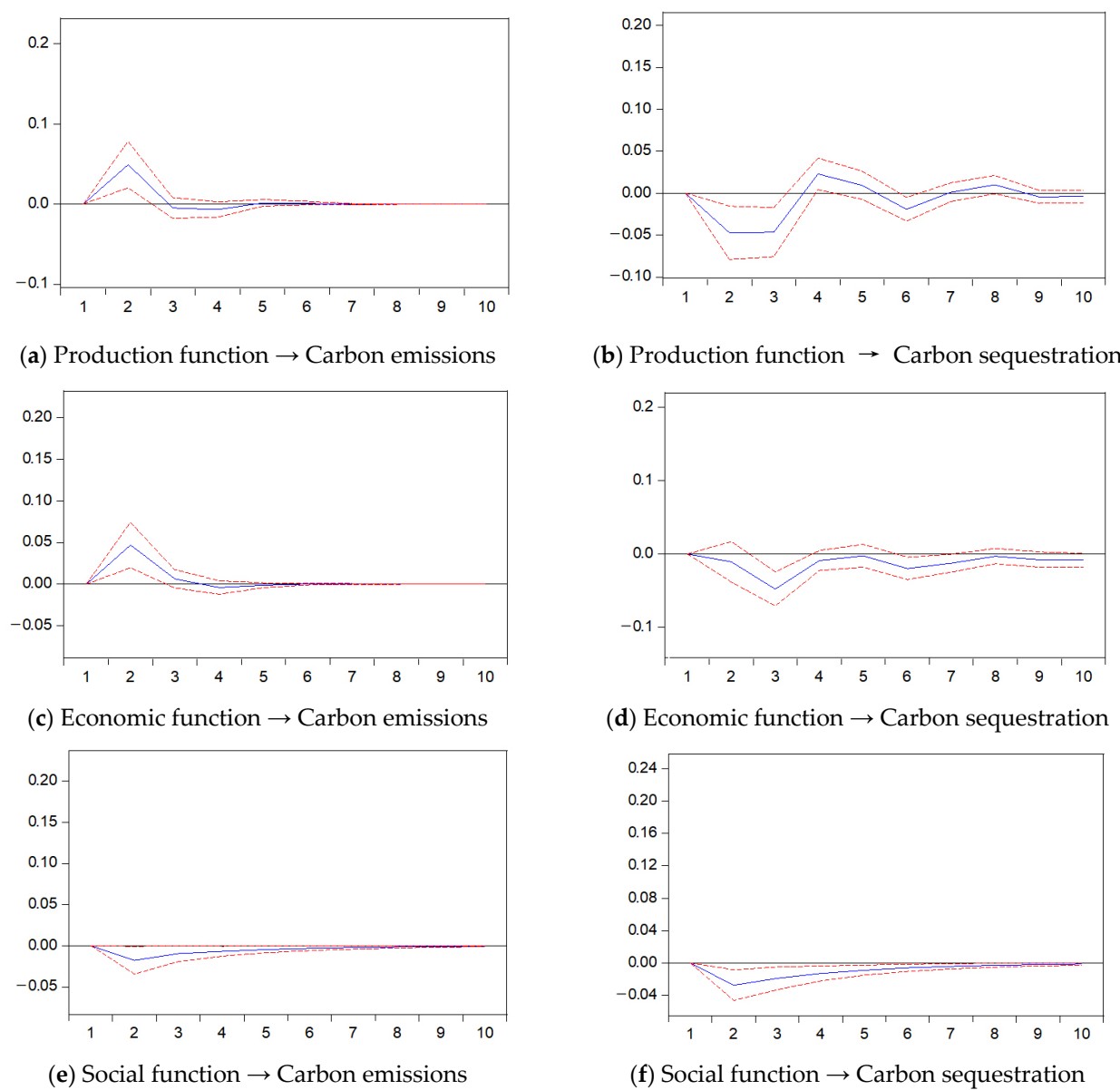

**Figure 12.** Impulse response functions of carbon effects and agricultural functions in the areas with strong agricultural functions.

The areas with strong agricultural functions located at the edge of the PRD have short-comings and bottlenecks in technical equipment, scale management, talent reserve, policy supports, etc., not realizing the intelligent and low-carbon development of the production function. From the perspective of agricultural technology, the innovative research and development ability of agricultural technology and equipment is not strong [60], and the technology application is difficult to adapt to the differentiated environment in the areas with strong agricultural functions. The utilization rate of agricultural resources is low, the reserve of green and low-carbon agricultural development technology is insufficient, and the supporting system is not perfect [56]. From the perspective of agricultural management, the smallholder management mode has a small scale, difficult financing, and low cultural quality in the areas with strong agricultural functions [61], which is difficult to form scale effects for and is not conducive to the promotion of low-carbon technologies, equipment, and management measures. From the perspective of talent reservation, low education level, insufficient informal education, and lack of systematic and diverse existing training in the marginal areas of the PRD make it difficult to popularize low-carbon agriculture. In

addition, there is a separation between theory and practice in personnel training in agriculture and forestry colleges, which restricts the promotion and application of low-carbon technologies [54].

The lack and backwardness of agricultural development goals, market forces, and organizational management methods in the areas with strong agricultural functions cause the economic functions to produce the effect of increased carbon emissions. The low-carbon development goal has not been fully taken into account in the top-level design of the agricultural development, and the agricultural economic development is at the expense of the environment, facing the dual constraints of resources and environment [62]. The market forces promoting the development of smart agriculture are insufficient. Both the integration of agriculture and the carbon trading market, and the participation of enterprises in agricultural development, are not high. The benefit linkage mechanism is not perfect. All of these make it difficult to ensure an increase in income even if low-carbon and intelligent production is achieved through the scale management of production services such as trusteeship [54].

According to the results of Granger causality test, the social function of the three agricultural regional types of the PRD have the carbon effect of emission reduction. From the perspective of the demonstration and the driving role of agricultural socialized services, social institutions are encouraged and guided to participate in agricultural socialized services under the goal of "double carbon" in the PRD. The joint co-operation between agricultural service subjects is strengthened, and the quality of agricultural socialization services is improved, introducing new technologies, new equipment, and new models that are both low-carbon and smart into the agricultural production of small farmers. Digital technology is used to monitor and evaluate the carbon reduction effects of socialized agricultural services, improving the credibility and transparency of socialized agricultural services. From the perspective of farmers' intentions and the behavior of low-carbon production, farmers in the PRD are environmentally conscious and active in learning about emission reduction. The more attention farmers pay to the ecological environment and safe production, the deeper the understanding of agricultural carbon emission reduction measures and effects they have, and the higher the environmental safety of their production behavior will be [63]. The resource allocation effect of market factors such as agricultural supply, agricultural product price, and market information will affect farmers' low-carbon production behaviors. The higher the risk-aversion degree, the more farmers are inclined to increase the input of agricultural materials to avoid the potential loss of output and income [64,65]. Social organizations have the natural attribute of connecting farmers, and the organizational advantage of solving the contradiction between small farmers and large markets [66]. Social services in the urban agriculture of the PRD play an important regulatory role in the implementation of agricultural carbon reduction measures.

According to the results of the Granger causality test, the three agricultural functions in the PRD have the carbon effect of sequestration reduction. On the one hand, the heavy reliance on petroleum and mineral resources, and the heavy investment of pesticides and fertilizers, are extremely unfriendly to the soil and the ecological environment, resulting in a serious decline in the quality of the cultivated land. A large amount of residual agricultural film is difficult to degrade, causing it to form a layer that is difficult to cultivate, and not easily permeable and breathable in the 15~20 cm soil layer. This not only affects the soil permeability, but also damages the soil diversity, which is not conducive to the organic balance of the ecological environment [67]. With the improvement of farmland management measures such as no tillage and less tillage, crop rotation, and straw return, the organic carbon content of cultivated soil has increased, but its organic carbon density is still relatively low compared to natural soil that has not been overutilized [68]. On the other hand, the expansion of cultivated land means that more natural forests, grasslands, and wetlands are reclaimed, leading to soil degradation, worsening the growth environment of food crops and other green vegetation, and significantly reducing the carbon sequestration capacity of food crops. In addition, the adjustment of the natural land will increase the

degree of land competition, resulting in a sharp drop in the content of organic matter and humus in the soil, and a decrease in the total carbon sink of the terrestrial ecosystem [69–71].

## 4. Discussion

### 4.1. The Evolution of Agricultural Production Types and Urban Food Security

Agricultural production types refer to the combination mode formed by the geographical scope with basically the same structure, nature, and characteristics of agricultural production [72]. The evolution of agricultural production types have a certain impact on urban food security. From the perspective of planting structure (Figure A1), agriculture in the PRD was dominated by the production function at first, and the proportion of the sown area of grain crops in the planting industry increased. Afterwards, with the transfer of agricultural comparative benefits, the proportion of the sown area of grain crops decreased, and the proportion of the sown area of economic crops increased. From the perspective of land use (Figure A2), the urban sprawl encroached on cultivated land and led to a decline in the proportion of the cultivated land area. Afterwards, with the intensive development of urban agriculture, small plots of cultivated land were merged and the occupation of cultivated land by ridges and weirs was avoided to reduce the waste of cultivated land. At the same time, the proposal of cultivated land red lines avoids the encroachment of urban construction land on cultivated land to a certain extent. Therefore, the cultivated land area increases slightly. From the perspective of cultivated land morphology, the fragmentation of cultivated land in the PRD first increased and then decreased (Figure A3). Due to the organization and management mode of small-scale family production and contract farming, cultivated land in the PRD has become fragmented and dispersed. Since around 2014, the intensive characteristics have promoted the development of the scale, mechanization, and modernization of urban agriculture in the PRD.

With regard to the comparative benefit transfer of the agricultural planting structure, the loss and fragmentation of cultivated land increased the grain risk in the PRD. The PRD is in a situation of extreme food insecurity, with its grain self-sufficiency rate declining from 48.86% in 2015 to 31.53% in 2020 (Figure A4), being heavily dependent on food imports from eastern, western and northern Guangdong Province and other provinces. Urban agriculture has the potential to enhance food security. The introduction of agricultural technology helps to increase grain production, and the advantages of the short transportation distance and the market access to perishable goods enrich the types of agricultural products [73]. Agricultural products can be sold or consumed directly, and therefore food losses in screening, processing, and storage can be avoided to a great extent [74]. Highly transparent food sources, as well as the standardization and branding of production processes, improve the quality of agricultural products, and the strict monitoring of fertilizers and pesticides, as well as the avoidance of pollution risks, make agricultural products more beneficial to human health [75].

### 4.2. Policy Enlightenment and Suggestions

In order to strengthen the agricultural carbon effects of emission reduction and sequestration increase in metropolitan areas, we put forward differentiated strategic suggestions based on the regional types of agricultural functions. From the perspective of agricultural carbon emission reduction, for the areas with weak agricultural functions, the developed modern technology and human wisdom at the center of urban agglomeration should be utilized to make up for the lack of traditional experience, realizing the efficient circulation and the use of data and information. Low-cost digital tools need to be integrated, increasing the precision and accuracy of analysis and decision making, and therefore reducing resource waste. The important platform of territorial spatial planning should be utilized for rational layout, improving urban living environments with limited agricultural land. Based on this, the agricultural low-carbon industrial chain and ecological chain can be built with a deep integration of science, technology, and agriculture, and the sharing of resources and energy information [76].

For the areas with medium agricultural functions, taking advantage of the coexistence of superior cultivated land resources and the relatively developed economy, the agricultural production structure should be optimized, and new agricultural production and operation modes such as green agriculture, circular agriculture, and low-carbon agriculture should be developed [55]. In the pilot carbon trading market [77], the responsibilities should be clarified, the low-carbon certification system should be specified, and reasonable carbon prices should be determined. Agro-ecological principles can be applied to co-ordinate the environment, sustainability, and production goals [78], ensuring economic benefits while reducing carbon emissions.

For the areas with strong agricultural functions, the farmers' concept of high-carbon production should be changed, and the concept of a low-carbon win–win needs to be popularized. The farmers also need appropriate subsidies to alleviate the economic pressure. Scientific experience based on evidence such as the over-application of nitrogen fertilizer to reduce production [79] needs to be taught, and local knowledge can be used to supplement technical contributions [80], avoiding agricultural activities at the expense of the environment. The agricultural scientific and technological talents need to be cultivated, the interaction with the market and enterprises needs to be strengthened, and the resource utilization needs to be improved.

For improving the soil's carbon sequestration capacity, the implementation of protective tillage systems, scientific agricultural management, and high-standard farmland construction needs to be strengthened. The continuous promotion of protective measures such as conservation tillage, straw return, intermittent irrigation in paddy fields, reasonable tillage rotation, manure application, clean energy instead of traditional energy, artificial grass planting, and diversified planting structures can effectively improve the organic matter content of farmland, cultivate the carbon sequestration capacity of farmland, and improve the ecological environment of farmland [76]. Additionally, ecological restoration based on plants and micro-organisms is mainly to use the growth and absorption of plants and micro-organisms themselves, or to indirectly remove pollutants. Certain micro-organisms can change the form and effectiveness of the elements in plant roots, enhance the adaptability of plants under heavy metal stress, and promote the absorption of heavy metals by plants. Phytoremediation is strengthened and the soil's carbon sequestration capacity is improved to achieve soil governance and carbon neutrality [81].

Land and water are important ecological factors limiting the development of urban agriculture. On the one hand, the lack of land resources can be solved by integrating the land and improving land-use efficiency. The land fragmentation in the areas with weak agricultural functions is serious, so it is necessary to improve the land-use efficiency with the support of science, technology, and capital. The areas with medium and strong agricultural functions should expand the scale of the agricultural land [82], forming large-scale production, and adopt the urban–rural integration development model to reclaim the land formerly used for residential land [83], so as to achieve the purpose of ensuring food security and protecting the environment. On the other hand, crop planting areas, seeding structures, and yield effects will cause changes in the water footprint [84], especially the increase in the water footprint and urban water consumption caused by agricultural expansion [85], which may lead to urban water conflicts. In response to the problem of water resources, the prevention and treatment of water pollution, the addition of inter-regional water transfer facilities, the reduction of the leakage of water infrastructure into the fields, the replacement of flood irrigation with spray or drip irrigation, and the better control of the time and place of irrigation [86] are ways to alleviate water conflicts.

### 4.3. The Boundaries of Green and Low-Carbon Transformation of Urban Agriculture

The innovation of agricultural production methods, the change in agricultural organization modes, the impact of market orientation, and the transfer of the agricultural labor force are decisive factors in the process of the green and low-carbon transformation of urban agriculture. However, it is necessary to take into account that the positive effect of

these factors on agricultural transformation is limited; that is, excessive input may impair the development of agriculture. The intensive agricultural production mode under the high input mode will lead to "agricultural involution" (that is, continuous additional investment per unit of land area results in a continuous reduction of marginal returns), failing to bring higher returns to farmers. The transfer of arable land caused by the large-scale mechanization of agricultural production and "de-peasantization" strategies [87] may lead to land grabbing, posing a threat to food security and social stability [88]. The market system in which farmers participate has gradually expanded from a single local production network to a complex urban and rural production network [89]. As the livelihood security system of farmers is gradually determined by the market, the cultivation of high value-added crops will inevitably bring more market risks. The large-scale and intensive development of agriculture has led to a large number of agricultural labor transfers, but the current improvement in the quality of agricultural labor cannot make up for the reduction in its quantity. High-quality and high-skilled labor is still lacking, which has become a limiting factor for agricultural transformation [90].

## 5. Conclusions

This paper innovatively combines urban agricultural multifunctionality with carbon effects. In terms of research methods, SOFM was used to classify agricultural regional types in the PRD according to the three major functions of urban agriculture, and Granger causality analysis was used to test the carbon effect of urban agricultural functions. Finally, the carbon effects of agricultural functions were analyzed based on the differences between the three agricultural regional types, considering the continuity and heterogeneity of spatio-temporal dimensions. The abundant data at the county scale for 19 consecutive years from 2002 to 2020 increase the robustness of the results of the index calculation and correlation test. The conclusions are as follows.

(1) The areas with strong basic agricultural functions are generally located at the edge with relatively backward development, and show a shrinking trend in scope, such as with the production function. The areas with strong intermediate agricultural functions are also distributed at the edge, but their scope is slowly expanding from the outside in, such as with the economic function. The areas with strong advanced agricultural functions such as the social function generally first appear in areas close to the core with a certain agricultural foundation and relatively developed socio-economic conditions, and the areas with strong advanced agricultural functions spread outward from relatively core areas.

(2) The PRD can be divided into three regions: the areas with weak agricultural functions, the areas with medium agricultural functions and the areas with strong agricultural functions. The reasons for the differences in the carbon effects produced by these different types of agricultural regions are related to multiple dimensions such as the agricultural ecological background, the agricultural production mode, agricultural operation and management, agricultural resource utilization, agricultural technology and talent reserve, the agricultural green and low-carbon industrial chain, government guarantee and market allocation, and agricultural socialized service.

(3) In the evolution of agricultural production types in the PRD, with regard to the comparative benefit transfer of agricultural planting structure, the loss and fragmentation of cultivated land increases the grain risk, and urban agriculture has potential in improving food security.

(4) Based on the regional types of agricultural functions and considering the constraints of land and water, strategic suggestions such as integrating natural resources, improving utilization efficiency, upgrading technical facilities, and avoiding production pollution are put forward.

(5) The green and low-carbon transformation of urban agriculture has its boundaries. The positive effects of the factors, namely the innovation of agricultural production

methods, the change in the agricultural organization modes, the impact of market orientation and the transfer of the agricultural labor force, are limited.

The findings of this paper have value and implications for academia, policy makers, producers, and ultimately for the local population in general. For academia, this paper deepens the theory of agricultural multifunctionality and provides a new perspective of the combination of agricultural function and carbon effects. For the decision makers, this paper provides innovative ways and theoretical support for the green development and regulation of urban agriculture, and the optimization and management of the ecological environment system. For producers, this paper strengthens the low-carbon concept and provides low-carbon production measures. Extending to the local population in general, the development of urban agriculture can improve the urban living environment, and the goal of carbon neutrality contributes to human well-being.

This paper also has its limitations. For the research on agricultural carbon effects, the carbon cycle process, and the mechanism of atmosphere, the phytosphere and pedosphere in the ecosystem are still not deeply explored. In the future, other agricultural regional types such as grassland, lake, and forest will be combined. Based on the theory of the regional system of the human–land relationship, we will explore the interaction and coupling relationship between human agricultural production activities and agro-ecological resources, and the environmental background and its impact on the carbon cycle, further strengthening the optimal management path of the urban agricultural regional system.

**Author Contributions:** Conceptualization, Z.S.; methodology, W.L.; formal analysis, Z.S.; investigation, Z.S.; data curation, F.L.; software, W.L.; writing—original draft preparation, Z.S.; writing—review and editing, J.Y.; visualization, Z.S.; supervision, J.Y.; project administration, J.Y.; resources, J.Y.; funding acquisition, J.Y. All authors have read and agreed to the published version of the manuscript.

**Funding:** This research was funded by the Natural Science Basic Research Plan of Shaanxi Province, grant number 2023-JC-YB-275.

**Data Availability Statement:** The data are not publicly available for privacy reasons.

**Acknowledgments:** Special thanks to Song Yongyong from Shaanxi Normal University for his guidance and revision of this article.

**Conflicts of Interest:** The authors declare no conflict of interest.

## Appendix A

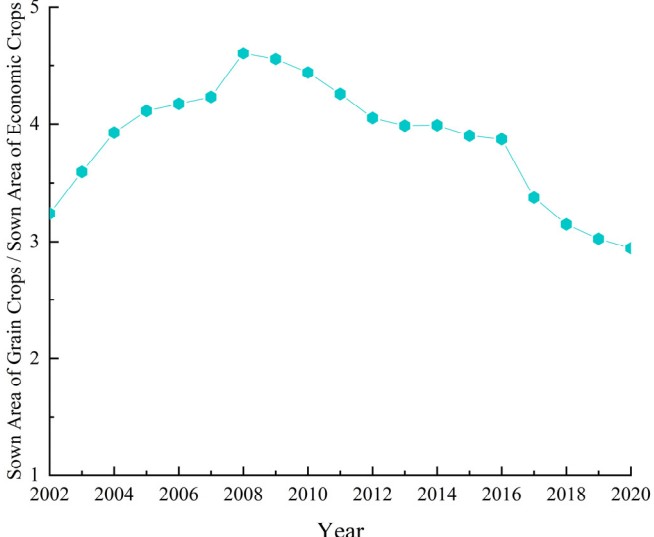

**Figure A1.** Ratio of sown area of grain crops to economic crops in PRD from 2002 to 2020.

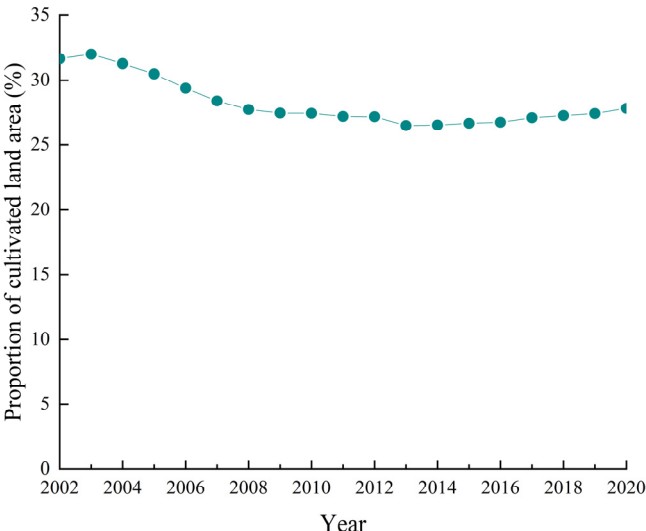

**Figure A2.** Proportion of cultivated land area in PRD from 2002 to 2020.

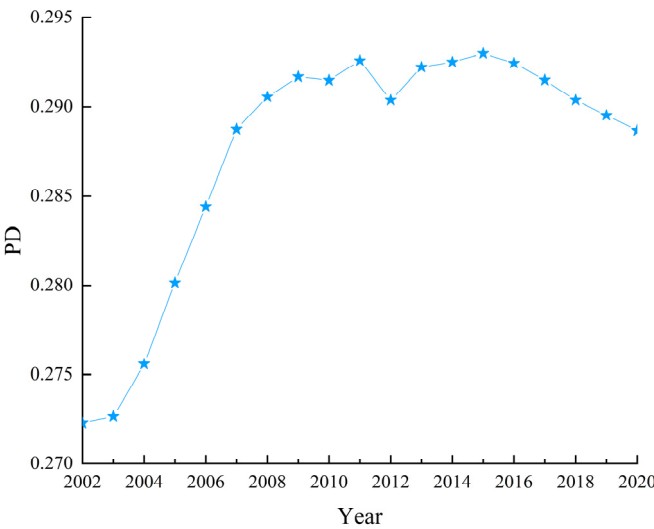

**Figure A3.** Patch density (PD) of cultivated land in PRD from 2002 to 2020.

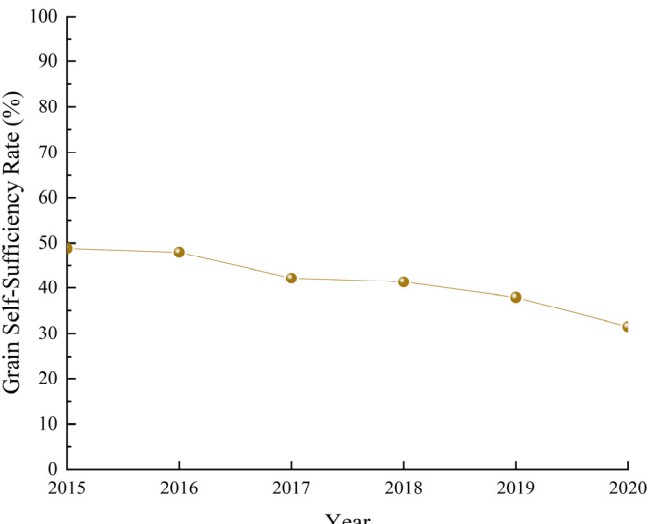

**Figure A4.** Grain self-sufficiency rate in PRD from 2015 to 2020.

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
