# Peer review of "Classification of Urban Agricultural Functional Regions and Their Carbon Effects at the County Level in the Pearl River Delta, China"

_agriculture, doi:10.3390/agriculture13091734_

Round 1

Reviewer 1 Report

The paper is interesting and certainly offers a novel vision to the topic. It can be however improved. In terms of the length and quality of sections the Abstract should be shortened and improved by presenting the relevance of the main findings of the paper (theoretically and for policy making). Section 4.6 repeats a summary of the main findings already presented elsewhere. Actually the last paragraph is the one that address what is expected in that section. Consider to erase all the previous text and expand the last paragraph considering the "content" suggestions offered bellow. Conclusions should not present a summary of the findings again. Instead one may expect to find a reflection of the authors around the meaning and use of their findings for academia, policy makers, producers and ultimately for local population in general.

Regarding the content, the piece presents three major shortcomings: (1) some calculations / modeling are not described properly. For example, the carbon calculations in section 3.2 need to be detailed as well as those related to the entropy weight method and the Granger causality. I strongly recommend to include those details as a supplementary material. (2) Major findings are a description of what the authors find by running their model (including the causalities presented) but there is no argumentation related to those findings. Do the findings and causalities established make any sense? How the authors can explain the tendencies find in different areas of the PRD? All this aspects are relevant to actually validate their findings. (3) Most of the solutions proposed for different agricultural functions are pretty much the same (mostly measures towards an intelligent agriculture heavily based in techno solutions). This aspect seems problematic as it suggest that there is no need of an analysis such as the one developed by the authors if solutions are going to be pretty much the same. Also it opens several questions such as what is the role of agroecology and low-cost solutions, what is the role of local knowledge, and mostly, what is the carbon footprint of implementing and ambitious policy that seeks to incorporate sensors and ITs technologies in urban agriculture...does this makes sense in terms of carbon emissions? is there a breaking point?

In addition to the above said, two aspects should be considered by the authors. On one hand, the type of agricultural production in the PRD (there were any meaningful changes during the analyzed period of time and if any, what they mean) and the relevance of such production for urban food security. If urban agriculture output represents a limited fraction of total urban consumption, this should be noted in order to reveal urban dependency to food importation and consequently the potential that urban agriculture may have in reducing such importations.

On the other hand, when making policy suggestions there is no consideration on two limited factors, land availability (and land footprint of different agricultural production schemes / techniques) and water availability (which is always a concern for urban settlements...if agriculture expands and water footprint of food increases, water conflicts may rise....how can the authors consider this issue in their policy suggestions?

Please consider boundaries of the proposed actions for green and low carbon transformation of urban agriculture in the PRD. If productivity and efficiency are some of the goals to pursue, how moving towards that end can change the main features of the proposed levels of agricultural functions? Chasing those goals, for example in the case of medium agricultural functions, could reach a point in which such level becomes strong agricultural functions with all their desirable and undesirable features? Why not or how that can be avoided?

Lastly, in lines 91-96 of page 2 add references that support the statement offered. In line 106, page 3, the statement offered can also be the opposite in some cases, please indicate so. In line 107, page 3, is argued that efficiency will lead to carbon emissions...yet efficiency is expected reduce carbon emissions but this is not what the text says. Perhaps the authors mean productivity instead of efficiency. Please revisit. In line 125-126, page 3, the argument seems repetitive (green and low carbon are features of sustainable development, isn't it?). Line 177, page 4, use income instead of " get rich".

Reviewer 2 Report

I found the paper interesting, the topic considers the three major function of urban agriculture and a Multifunctional Index System for Urban agriculture at regional level. I found appropriate the use of Granger test. But the paper could be improved. I found the paper too long: In my opinion you should try to summarize some sections, starting from the Abstract. The Discussion is good, but too broad, the reader may lose the thread. Furthermore, please consider to moving some of the consideration made in paragraph 4.6 to Conclusions (that should be revised as well). In the Conclusions the objective and results are recalled again. I would opt for short conclusions that includes limits and policy implication (so please consider to move policy implication to Conclusion as well, but in a nutshell).

I suggest providing a few more references about the causality test in paragraph 2.3.4, starting from Granger studies (1969) to Sims (1980), Hamilton (1994) and Stock&Watson (2009). This will allow a better explanation on the VAR models, the causality concepts and hypothesis test. Then results will be better addressed. In line 267 you said “…became obvious”. Please specify. 
